# THE DEVIL WEARS TOKENS: TOKEN-LEVEL ATTENTION AND ALIGNMENT ANALYSIS UNCOVERS HALLUCINATIONS IN LVLMS

## ABSTRACT

Large vision-language models (LVLMs) achieve strong results in visual reasoning and question answering but remain vulnerable to hallucination. Previous studies of hallucination have focused on global statistics and image-level attention. We present instead a fine-grained analysis of object hallucination in LVLMs at the patch-token level across layers; and show that token-level statistics is more reliable to detect hallucination than image-level statistics. Our study reveals two core findings: **(i)** hallucinated tokens exhibit diffuse, non-localized attention, while faithful tokens show compact attention on relevant patches; and **(ii)** hallucinated text tokens are not aligned with any object regions. Leveraging these insights, we introduce a lightweight, explainable hallucination detector based on patch-level statistics and hidden features, achieving up to 90% accuracy in token-level hallucination detection. These results demonstrate the value of structural, token-level analysis for understanding and mitigating hallucinations in LVLMs.

## 1 INTRODUCTION

Large Vision-Language Models (LVLMs) such as LLaVA Liu et al. (2023), InstructBLIP Dai et al. (2023), Otter Li et al. (2023a), mPLUG-Owl Ye et al. (2023), MiniGPT-4 Zhu et al. (2023), and Qwen-VL Bai et al. (2023) have achieved remarkable progress across multimodal tasks, including captioning, visual question answering (VQA), and dialogue. However, these models are vulnerable to a critical failure mode: *object hallucination*, where the model generates descriptions of objects that are not present in the image Rohrbach et al. (2018); Li et al. (2023b). Such hallucinations undermine reliability and safety, which can be a problem in domains such as robotics and medicine.

While prior works Chen et al. (2024); Leng et al. (2024); Chuang et al. (2024); Huang et al. (2024) have made important strides in benchmarking and mitigating hallucinations, most analyses remain at a *coarse-grained* and *global* level. They examine aggregated global statistics, such as the total attention assigned to relevant patches Chuang et al. (2024), or the attention weights of preceding tokens Fieback et al. (2024). These approaches Fieback et al. (2024); Jiang et al. (2025); Yang et al. (2024) provide useful signals but fail to capture the interactions between patch tokens inside LVLMs. As a result, the internal mechanisms of hallucination - how attention distributions, hidden states, and token-wise activations behave during hallucination - remain unexplored. A comparison of our method with other object hallucination detection methods is displayed in Fig. 1 As such, current models mis-detect hallucinations when encountered with hallucinated objects that are "similar" in semantic meaning compared with other true objects, which can be evidenced in Figure 5. In other words, there is a critical gap in *token-level analysis*, which can reveal the internal divergences between hallucinated and faithful object tokens.

Our work addresses this limitation through a fine-grained investigation of hallucination behaviors via *token-level interactions*, displayed by Fig. 1. By analyzing cross-modal patch-level attention maps and hidden state activations, we demonstrate that hallucinated object tokens exhibit distinct internal signatures compared to tokens grounded in real objects. In particular, our study reveals two findings. **1)**, hallucinated tokens exhibit diffused and scattered attention. **2)**, hallucinated tokens have low semantic similarities with *any* image regions. These findings are illustrated in Fig. 2

Figure 1: *Left:* Comparison with three existing methods for Object Hallucination Detection, which only analyze **global statistics**. HalLoc Park et al. (2025) uses a black-box VisualBERT (yellow block) to model hidden visual and textual features. MetaToken Fieback et al. (2024) computes first-order token probability statistics (orange block) as hallucination indicator. SVAR Jiang et al. (2025) measures global attention sum on visual tokens (blue block). In contrast, our study studies **structural behaviors** of patch tokens via *interpretable* feature similarities and patch-wise spatial attention entropy (green block). *Right:* existing methods summarizes the attention and statistics globally. When some noisy tokens have peak attention, the global statistics also peaks. In contrast, we analyzes the structured and fine-grained statistics on the *patch* level, thus successfully detecting hallucinations in complex scenes.

and 5. Finally, we show that our discovered behaviors yield state-of-the-art hallucination detection accuracy compared with existing works.

Our contributions are threefold:

- We introduce novel techniques to discover the *structural behaviors* of LVLM hidden layers when hallucinating, leveraging patch-level attention distributions and internal feature representations. As a result, we derive two intuitive statistics to measure LVLM's hallucinatory behavior: *Patch-wise Attention Spatial Entropy* and *Patch-wise Cross-Modal Feature Similarities*.

- **Key findings**: First, compact and well-localized attention distributions are strongly associated with real objects, whereas hallucinated tokens exhibit diffuse and scattered attention over image patches. Second, non-hallucinated tokens have high similarities with the mentioned image regions, while hallucinated tokens show weak or inconsistent semantic alignment with visual patches. Together, these results indicate that hallucinations are primarily driven by language priors rather than deficiencies in the vision encoder.

- **Detection framework**: Building on these insights, we propose a simple yet effective token-level hallucination detection method that combines hidden representations with statistical features such as entropy, patch similarity, and confidence alignment. Our framework achieved a state-of-the-art detection accuracy of up to 90 %, demonstrating the effectiveness of our analysis methodologies and validity of our findings.

These findings demonstrate that local patch-level characteristics accurately reflect how well the model truly knows an object, offering a principled basis for internal, token-level hallucination detection.

## 2 RELATED WORKS

### 2.1 LARGE VISION-LANGUAGE MODELS

Large Vision-Language Models (LVLMs) have achieved impressive results across a wide range of multimodal tasks, including captioning, visual question answering (VQA), and instruction following. Representative systems include LLaVA Liu et al. (2023), InstructBLIP Dai et al. (2023), Flamingo Alayrac et al. (2022), Otter Li et al. (2023a), MiniGPT-4 Zhu et al. (2023), PaLI Chen et al. (2023), Qwen-VL Bai et al. (2023), and Gemini DeepMind (2023). These models integrate large-scale vision encoders with instruction-tuned LLMs, enabling flexible multimodal reasoning and generation.

However, despite their success, LVLMs remain vulnerable to object hallucination, where the model incorrectly asserts the presence of non-existent objects Li et al. (2023b); He et al. (2025). Such failures undermine reliability in safety-critical settings. Unlike prior work that mainly reports these limitations, our study investigates the internal, patch-level mechanisms that underlie hallucinations.

### 2.2 OBJECT HALLUCINATION BENCHMARKS

To measure hallucination, several benchmarks have been proposed. In image captioning, Rohrbach et al. (2018) introduced CHAIR, a metric that quantifies hallucinated object mentions. More recently, Li et al. (2023b) developed POPE, which evaluates a model's ability to deny hallucinated objects in binary prompting setups. He et al. (2025) erased objects from images to test persistence of false mentions, and Chen et al. (2024) proposed ROPE, a multi-object benchmark showing that hallucinations become more prevalent in complex scenes.

While these benchmarks are valuable, they evaluate hallucination at the global, image-level by measuring model outputs. They do not capture the local, structural dynamics within the model that give rise to hallucination. Our work instead focuses on patch- and token-level statistics, which reveal the internal mechanisms behind these benchmarked behaviors.

### 2.3 LVLM HALLUCINATION ANALYSIS

Beyond benchmarks, recent work has begun examining the model internals. Yang et al. (2024) performed token-level attribution and entropy analysis, showing that hallucinated tokens attend less to relevant patches. Gong et al. (2024) exploited token-level attention to guide generation, improving over decoding-based mitigation methods such as VCD Leng et al. (2024). Zhou et al. (2024) traced hallucinations to mid- and high-layer attention heads that over-rely on language priors, demonstrating that reweighting these "hallucination heads" can reduce false objects.

These studies highlight attention's role in hallucination but remain limited to coarse or global measures, such as total attention mass or attribution scores. In contrast, our work provides a structural, patch-level analysis of both attention and hidden features, uncovering fine-grained signals that distinguish hallucinated from faithful tokens.

### 2.4 HALLUCINATION DETECTION

Several approaches detect hallucinations at the output level. LogicCheckGPT Wu et al. (2024) probes logical consistency across related queries, while Woodpecker Yin et al. (2024) validates object mentions using external vision models. Recent methods such as HSA-DPO Xiao et al. (2025) and M-HalDetect Gunjal et al. (2024) employ proprietary models to flag hallucinations at the sentence level. More fine-grained efforts include MetaToken Fieback et al. (2024), which detects token-level hallucinations using internal states, and HalLoc Park et al. (2025), which introduces a benchmark and detection framework for diverse hallucination types .

These methods are effective but treat LVLMs largely as black boxes, focusing on verification or classifier-based detection. Our approach, in contrast, derives detection features directly from local structural signals—including patch-level attention and hidden feature coherence—providing both interpretability and practical improvements in token-level hallucination detection.

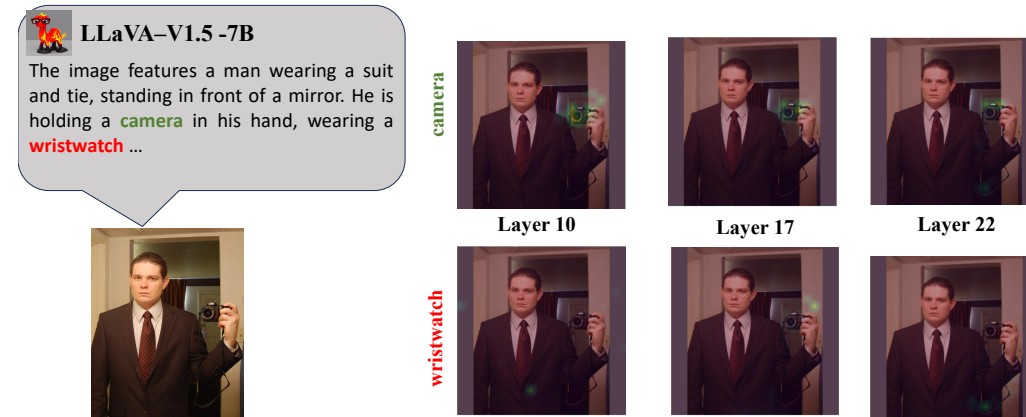

Figure 2: The visual attention maps for or a true token ("camera", top) vs. a hallucinated token ("wristwatch", bottom) across layers (10, 17, 22). True tokens reveal a focused attention pattern aligned with the object's location, while hallucination tokens have scattered attention across the image.

## 3 TOKEN-LEVEL ANALYSIS OF LARGE VISION-LANGUAGE MODELS

We conduct token-level investigation, revealing the distinct interactions between hallucinated tokens and the image patch tokens, thus, useful to detect hallucination. Our study yields two key findings: hallucinated tokens (i) have diffused attention, and (ii) show low grounding with any image patches. To quantify the two behaviors, we introduce two **patch-level structural statistics**: (i) attention spatial entropy, and (ii) patch-level feature similarities. We also find a minor finding: our proposed statistics has positive correlations with the model confidence.

### 3.1 SETUP

We report results on a range of popular open-source LVLMs: LLaVA-V1.5-7B, Qwen2.5-VL-8B, MiniGPT-v2, and Llama-3.2-11B-Vision. Models are prompted with *"Describe this image."* and decoded greedily unless otherwise stated. We curate a dataset of 4000 MS-COCO 2014 images from the validation set with a 90/10 split and classify generated object tokens as either True Object or Hallucinated Object using GPT-4o.

### 3.2 ATTENTION SPATIAL ENTROPY

**Discovery.** LVLMs exhibits higher *scattered* attention distributions over patches when producing hallucinated tokens, whereas correct tokens display *compact* and *localized* attention. Intuitively, we can look at the following qualitative example in Fig 2, where we can see the model is able to allocate enhanced attention to a focused area if the object is not hallucinatory; while failing to do so in hallucinated objects.

**Spatial entropy reveals hallucinated tokens.** To validate the hypothesis, we propose attention entropy to model the distribution of the attention map, and quantify the *compactness* of attention distribution. For a generated token $t$ at layer $n$, let $\mathbf{A}_t^{(n,h)} \in \mathbb{R}^{|\mathcal{P}|}$ denote the attention from head $h$ over the patch set $\mathcal{P}$ (e.g., a $24 \times 24$ patch grid). We first average across $H$ heads to obtain a layer-wise attention map over visual patches:

$$\bar{\mathbf{A}}_t^{(n)} = \frac{1}{H} \sum_{h=1}^{H} \mathbf{A}_t^{(n,h)}, \qquad \bar{\mathbf{A}}_t^{(n)}(p) \geq 0, \quad \sum_{p \in \mathcal{P}} \bar{\mathbf{A}}_t^{(n)}(p) = 1.$$

To isolate the most salient activations, we retain only the top-$x\%$ of patch responses (empirically $x = 10$ yield the most favorable results for us). The retained patches are grouped into 8-connected

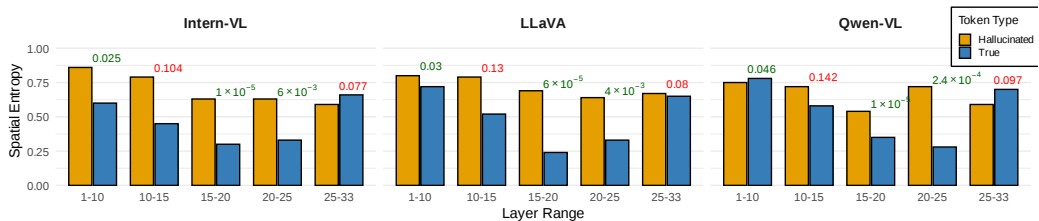

Figure 3: Layer-wise attention entropy of *true* vs. *hallucinated* tokens across LVLMs (lower is better; more focused). Reported $p$-values indicate strong separation in early/mid layers.

components $\mathcal{C}_t^{(n)}$, and small spurious blobs (area $< \theta$) are suppressed to filter out attention sinks. We apply a *quantile suppression* to preserve the attention structure, discarding attention sinks. Specifically, for patches that exhibit high attention but do not belong to any valid component, their value is replaced with the *median* of the original map. This produces an attention sink-suppressed but still structurally faithful distribution

$$\hat{\mathbf{A}}_t^{(n)}(p) = \begin{cases} \bar{\mathbf{A}}_t^{(n)}(p), & p \in \cup_{c \in \mathcal{C}_t^{(n)}} c, \\ \text{median}\left(\bar{\mathbf{A}}_t^{(n)}\right), & \text{otherwise,} \end{cases}$$

which is then renormalized to ensure all attention weights sum to one.

The *spatial compactness* of attention is quantified over the full renormalized map via Shannon entropy:

$$H_t^{(n)} = -\sum_{p \in \mathcal{P}} \hat{\mathbf{A}}_t^{(n)}(p) \log \hat{\mathbf{A}}_t^{(n)}(p).$$

Low entropy indicates that attention is concentrated in one or two contiguous regions (compact, grounded focus), while high entropy indicates scattered, diffuse activation across multiple disjoint regions with strong residual sinks.

Since attention sharpness varies across layers, we obtain a robust token-level score by averaging the $k$ most entropic layers:

$$H_t = \frac{1}{k} \sum_{n \in \mathcal{T}_k} H_t^{(n)},$$

where $\mathcal{T}_k$ indexes the top-$k$ layers ranked by entropy (empirically $k = 10$). We visualize the formula for our spatial attention entropy with Fig. 8.

**Remarks.** As shown in Fig. 3, grounded tokens consistently display lower entropy than hallucinations in early and mid layers, reflecting stronger visual focus before linguistic priors dominate at depth. Averaging the top-5 most entropic layers further stabilizes this separation across LVLMs.

**Spatial entropy as a hallucination detector.** With threshold $\tau_H$, hallucination detection follows:

$$\text{Halluc}_t = \begin{cases} 0 & \text{if } g_t \leq \tau_H \quad \text{(grounded)}, \\ 1 & \text{if } g_t > \tau_H \quad \text{(hallucinated)}. \end{cases}$$

Thus, grounded tokens are characterized by compact, low-entropy attention, while hallucinations exhibit high-entropy, spatially scattered attention.

**Results.** The attention-entropy indicator shows clear distributional separation and solid threshold-free performance across models (Fig. 4). Quantitatively (Table 4) displays that F1 scores lie in 0.73–0.80, establishing entropy as an informative *spatial compactness* signal that is complementary, but generally weaker than *feature alignment* (Sec. 3.3).

| Model | Prec. | Rec. | F1 |
|---|---|---|---|
| LLaVA-1.5-7B | 0.81 | 0.74 | 0.77 |
| MiniGPT-4 | 0.79 | 0.69 | 0.73 |
| Qwen-VL | 0.82 | 0.77 | 0.80 |
| LLaMA-3.1-V | 0.80 | 0.71 | 0.75 |

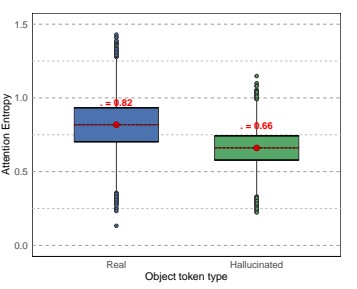 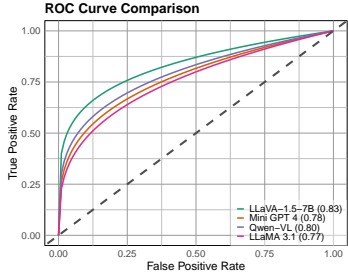

(a) Performance across LVLMs.  (b) Attention entropy for *true* vs.  (c) ROC curves across LVLMs.
*hallucinated* tokens.

Figure 4: Attention-entropy hallucination indicator: (a) tabular performance, (b) distributional separation, and (c) threshold-agnostic ROC curves.

## 3.3 VISUAL ALIGNMENT SCORE

**Discovery.** Correct tokens that correspond to visually present objects exhibit high peaks in patch-level similarity, while hallucinated tokens show low similarity over the image grid (Fig. 5). In other words, the predicted tokens have low alignment scores with visual patches are more likely to be hallucinated. This supports the view that the vision backbone preserves *structural information*, evidenced by LVLM's true object token's features to correspond positively to their actual location. This suggests that many hallucinations originate from *language priors* Park et al. (2025); Rohrbach et al. (2018).

This complements the attention-entropy finding: entropy captures *structural compactness* of the attention mass in image space, while similarity (Sec. 3.3) captures *structural alignment* between token and patch features in representation space.

**Patch-level alignment scores reveal hallucinations.** We evaluate the feature similarities between the hallucinated token and image patches; and coin this metric as patch-level alignment scores. At layer $n$, let $h_t^{(n)}, v_p^{(n)} \in \mathbb{R}^d$ be the token and patch embeddings, and define cosine similarity

$$S_{t,p}^{(n)} = \frac{\langle h_t^{(n)}, v_p^{(n)} \rangle}{\|h_t^{(n)}\|_2 \|v_p^{(n)}\|_2}, \tag{1}$$

which reflects *local structural alignment*. The per-token map is $\mathbf{S}_t^{(n)} = [S_{t,p}^{(n)}]_{p \in \mathcal{P}}$. To emphasize localized evidence, we obtain the token grounding score $g$ by aggregating the top-$k$ patches $\mathcal{T}_t^{(n)}$:

$$m_t^{(n)} = \frac{1}{k} \sum_{p \in \mathcal{T}_t^{(n)}} S_{t,p}^{(n)}. \tag{2}$$

Because alignment varies with depth, we select the most supportive layer $n_t^* = \arg\max_n m_t^{(n)}$ and use the highest patch score as the alignment score $m_t^{(n_t^*)}$.

**Alignment scores as the hallucination detector.** We classify tokens by thresholding the score:

$$\text{Halluc}_t = \begin{cases} 0 & \text{if } m_t^{(n_t^*)} \geq \tau, \\ 1 & \text{if } m_t^{(n_t^*)} < \tau. \end{cases}$$

To avoid arbitrary choices, $\tau$ is selected on validation data to maximize the F1 score.

**Qualitative and quantitative results.** Similarity maps (Fig. 5) reveal compact peaks for grounded tokens and dispersed responses for hallucinations. Layer-wise analyses across LVLMs (Fig. 6) show strong separation in early/mid layers: true tokens average $\approx 0.8$ vs. hallucinations $\approx 0.4$–$0.5$, with significance confirmed by $p$-values. The gap narrows after $\sim$layer 25, consistent with language-prior dominance at depth.

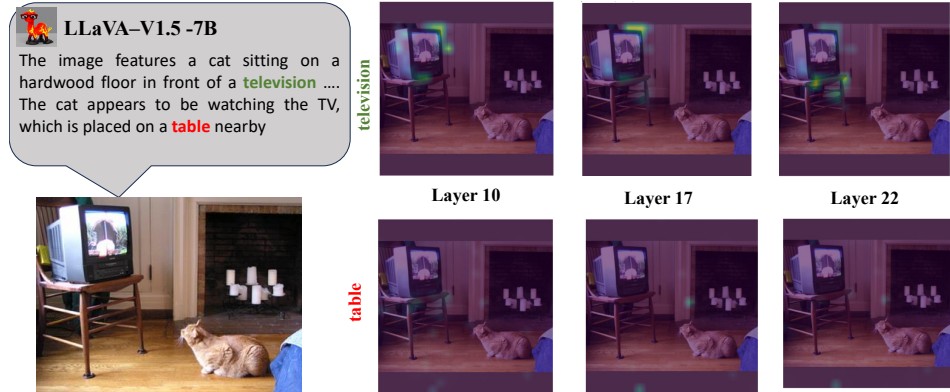

Figure 5: Feature-similarity heatmaps for a true token ("television", top) vs. a hallucinated token ("table", bottom) across layers (10, 17, 22). True tokens display similarity clusters aligned with the object's local structure, while hallucination tokens have low alignment with all patches across the image.

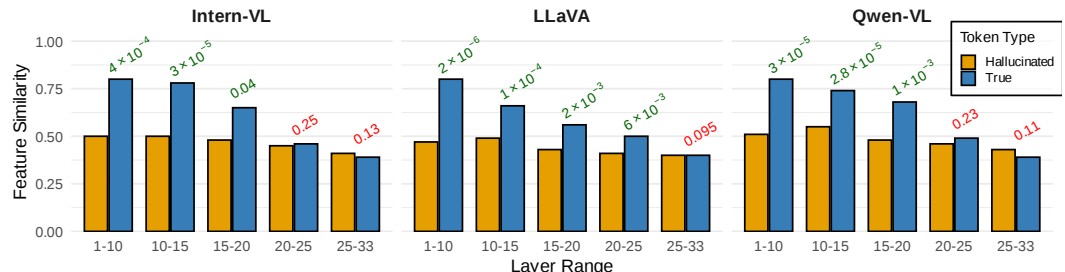

Figure 6: Layer-wise patch-level similarity scores for true vs. hallucinated tokens across LVLMs. Early/mid layers are most discriminative; deeper layers converge due to stronger language priors.

**Indicator performance.** A simple threshold on top-$k$ similarity delivers strong detection accuracy across models, as shown in Table 5; Fig. 6. This establishes local feature similarity as a powerful *structural alignment* signal, complementary to attention-entropy's *spatial compactness* signal.

## 4 TOKEN-LEVEL HALLUCINATION DETECTION AND EVALUATION

### 4.1 DETECTOR'S FEATURES

To show the effectiveness of our discovered hallucination indicators, we apply a lightweight classification model using the indicator features. Our approach is deliberately straightforward yet interpretable, relying on our two proposed indicators:

1. **Spatial Attention Entropy** (§3.2) — measures the spatial compactness of attention maps associated with an object token, after applying $k$-connected component filtering ($k = 8$) to suppress noisy isolated activations. Tokens grounded in the image typically show compact, low-entropy attention distributions, whereas hallucinations yield dispersed, high-entropy activations.

2. **Patch-level Alignment Score** (§3.3) — quantifies the top-$k$ cosine similarity between a token embedding and visual patch embeddings, capturing structural alignment between the linguistic representation and localized visual evidence. True tokens generally yield sharp similarity peaks over relevant patches, while hallucinations display weak or diffuse similarities.

| Model | Prec. | Rec. | F1 |
|---|---|---|---|
| LLaVA-1.5-7B | 0.82 | 0.76 | 0.79 |
| MiniGPT-4 | 0.80 | 0.73 | 0.76 |
| Qwen-VL | 0.83 | 0.79 | 0.81 |
| LLaMA-3.1-V | 0.80 | 0.71 | 0.75 |

(a) Performance across LVLMs.  (b) Top–5% similarities for *true* vs. *hallucinated* tokens.  (c) ROC curves across LVLMs.

Figure 7: Similarity-based hallucination indicator: (a) tabular performance, (b) distributional separation, and (c) threshold-agnostic ROC curves.

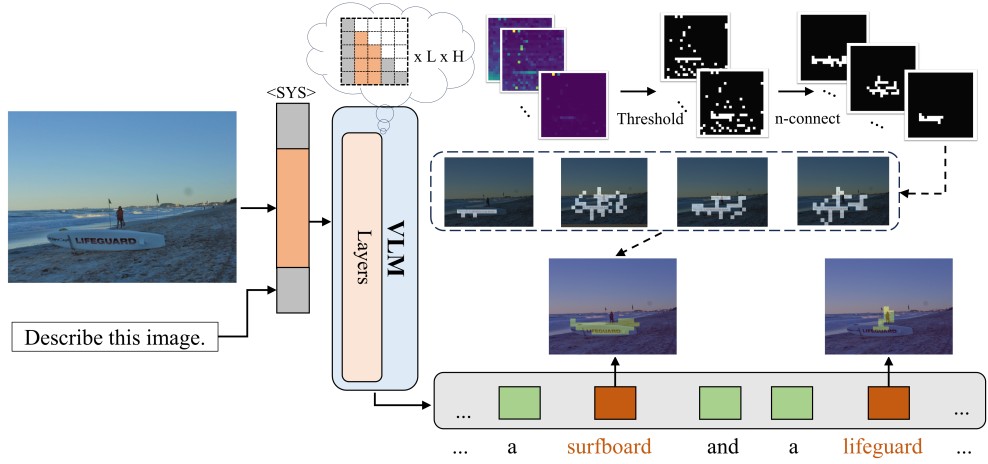

Figure 8: Illustration of Spatial Entropy calculation, with top-k components and thresholding

For each token $t$ generated by a LVLM, we compute both measures across all transformer layers $n \in \{1, \ldots, L\}$. Denote the per-layer entropy values as

$$\mathbf{H}_t = [H_t^{(1)}, H_t^{(2)}, \ldots, H_t^{(L)}],$$

and the per-layer similarity scores as

$$\mathbf{M}_t = [m_t^{(1)}, m_t^{(2)}, \ldots, m_t^{(L)}].$$

The concatenated feature vector

$$\mathbf{f}_t = [\mathbf{H}_t \,||\, \mathbf{M}_t] \in \mathbb{R}^{2L}$$

thus encodes both structural attention information and local feature alignment for token $t$.

We then train supervised classifiers to predict whether token $t$ is *grounded* or *hallucinated*, using $\mathbf{f}_t$ as input. We evaluate three model families:

- **Extreme Gradient Boosting (XGB)** — efficient at capturing non-linear interactions between entropy and similarity features.
- **Multi-Layer Perceptron (MLP)** — a shallow neural model that flexibly captures higher-order feature correlations.
- **Random Forest (RF)** — an ensemble of decision trees that is robust to noise and does not require feature scaling.

Each classifier is trained separately for each LVLM, as layer dynamics and attention–similarity patterns vary by architecture.

Table 1: Performance of our token-level hallucination detector across LVLMs.

| LVLM | Method | PR | RC | F1 | ACC |
|---|---|---|---|---|---|
| LLaVA | XGB | 0.87 | 0.83 | 0.85 | 0.915 |
| | MLP | 0.85 | 0.81 | 0.83 | 0.908 |
| | RF | 0.84 | 0.79 | 0.81 | 0.902 |
| InternVL | XGB | 0.88 | 0.84 | 0.86 | 0.921 |
| | MLP | 0.86 | 0.81 | 0.83 | 0.911 |
| | RF | 0.84 | 0.80 | 0.82 | 0.905 |
| MiniGPT-4 | XGB | 0.85 | 0.80 | 0.82 | 0.901 |
| | MLP | 0.83 | 0.78 | 0.80 | 0.894 |
| | RF | 0.81 | 0.77 | 0.79 | 0.888 |
| GPT-4V | XGB | 0.89 | 0.85 | 0.87 | 0.926 |
| | MLP | 0.87 | 0.83 | 0.85 | 0.918 |
| | RF | 0.85 | 0.81 | 0.83 | 0.910 |

Table 2: Comparison of token-level hallucination detection methods on LLaVA.

| Method | PR | RC | F1 | ACC |
|---|---|---|---|---|
| Token Entropy | 0.29 | 0.51 | 0.37 | 0.74 |
| Token NLL | 0.29 | 0.43 | 0.35 | 0.76 |
| HalLoc | 0.61 | 0.62 | 0.61 | 0.68 |
| MetaToken + LR | 0.68 | 0.23 | 0.34 | 0.87 |
| MetaToken + GB | 0.65 | 0.41 | 0.50 | 0.88 |
| SVAR | 0.68 | 0.71 | 0.69 | 0.80 |
| **Ours + XGB** | **0.87** | **0.83** | **0.85** | **0.91** |
| **Ours + MLP** | **0.85** | **0.81** | **0.83** | **0.91** |
| **Ours + RF** | **0.84** | **0.79** | **0.81** | **0.90** |

## 4.2 EXPERIMENTAL SETUP

We benchmark our detector against state-of-the-art baselines for token-level hallucination detection, namely MetaToken Fieback et al. (2024), HalLoc Park et al. (2025), and SVAR Jiang et al. (2025). Implementation details are presented in Supplementary [refer to which Section number].

For our method, the concatenated feature vector $\mathbf{f}_t$ (which combines per-layer spatial entropy and feature similarities) serves as input. XGB, MLP, and RF classifiers are trained on the training set, and tuned on validation data to save the best checkpoint.

## 4.3 RESULTS

**Across-LVLM Performance.** Table 1 reports results on four representative LVLMs: LLaVA, InternVL, MiniGPT-4, and GPT-4V. Our detector achieves consistent improvements across models, with GPT-4V and InternVL achieving the highest scores. MiniGPT-4 and LLaVA yield slightly lower recall, suggesting that their hallucinations may be more subtle, though entropy and similarity features still provide strong predictive signals.

**Comparison with Baselines (LLaVA).** Table 2 compares our detector with MetaToken, HalLoc, and SVAR using LLaVA. Our method substantially outperforms all baselines in F1, with XGB achieving the best overall balance between precision and recall. MetaToken exhibits relatively high precision but extremely poor recall, missing many hallucinations. HalLoc achieves balanced but weaker scores, while SVAR performs competitively but still trails behind our approach.

## 5 CONCLUSION

We presented a token-level, structural analysis of object hallucinations in LVLMs and turned the resulting insights into a lightweight, interpretable detector. Two simple statistics: Patch-wise Attention Spatial Entropy and Patch-wise Cross-Modal Feature Similarity, are shown to be especially diagnostic: hallucinated tokens tend to exhibit diffuse, non-localized attention and weak alignment to any image region, whereas faithful tokens show compact attention and strong, localized similarity to the relevant patches. Building on these findings, our detector, trained on per-layer entropy and similarity features, achieves strong, cross-model performance and outperforms prior token-level baselines significantly. These improvements arrive without heavy additional modules, providing a better path to hallucination detection in real systems.

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

## A EXPERIMENTAL CONFIGURATIONS

For hallucination detection, our configurations are as follows:

- **MetaToken**: For our greedy decoding setup, the probability difference term in Eq. 11 of the original paper is always zero; we instead use token probability directly. We implement two binary classifiers, Logistic Regression (LR) with `lbfgs` solver and Gradient Boosting (GB) with 100 estimators.
- **HalLoc**: We use a pretrained CLIP-ViT-B/32 encoder with a linear projection to match VisualBERT input dimensions. A single classification head is used, as we focus exclusively on object hallucination. Training follows the original optimizer setup (AdamW, $\beta = (0.9, 0.999)$, weight decay $1.0 \times 10^{-2}$, learning rate $1.0 \times 10^{-6}$) with batch size 16 on an RTX 3090.
- **SVAR**: We train a one-hidden-layer MLP (hidden dim 248, learning rate 0.001) for 50 epochs as a result of the hyperparameter search strategy.

For decoding strategies, our configurations are as follows:

- **Beam search**: num_beams = 5
- **VCD**: noise_step = 700, $\alpha = 1.0, \beta = 0.2$
- **OPERA**: num_beams = 5, $\sigma = 50$, $r = 15.0$, num_candidates = 5
- **AGLA**: $\alpha = 2.0, \beta = 0.5$

## B  DATASET CURATION

In order to construct a hallucination detection dataset, instead of following previous papers to use CHAIR Rohrbach et al. (2018), we used GPT-4o API to extract hallucinated words. This is because the CHAIR toolkit often misses or return excessive words in case of ambiguity. The prompt structure we used is as follows:

---

**Prompt**

You are given:
- A list of ground truth object classes (from COCO).
- A detailed description of an image.
- Several captions of the same image.

Your task:
Find all object classes that are mentioned in the description, but are NOT mentioned in any of the captions, and are NOT present in the ground truth list.
Output the result as a list as in the examples. Do NOT add any extra text or provide any explanations.

**Examples:**

*Objects:* ["bowl", "broccoli", "carrot"]
*Description:* There are two bowls of food, one containing a mix of vegetables, such as broccoli and carrots, and the other containing meat. *Captions:*
- A bowl with broccoli and carrots.
→ Output: ["meat"]

*Objects:* ["bowl", "broccoli"]
*Description:* - A bowl full of broccoli.
*Captions:* - A bowl of green vegetables.
→ Output: []

**Now answer:**
Objects: {objects}
Description: {description}
Captions: {captions_formatted}
→ Output:

---

It is possible that an object word can consist of more than 1 token, therefore we calculate the token-level attribute by indexing the object word, and obtain a 'prefix' which is the sentence prior to that word. Then the VLMs will forward once based on the 'prefix', yielding the internal attributes of the object token.

