# OpenReview forum: "The Devil is in the Tokens: Token-Level Structural Analysis Uncovers Hallucinations in LVLMs"
_ICLR.cc/2026/Conference — ICLR 2026 Conference Withdrawn Submission_

### Official Review · Reviewer_36ng · 2025-10-30

**Soundness:** 3
**Presentation:** 3
**Contribution:** 2
**Rating:** 4
**Confidence:** 4

**Summary:**

The paper presents a detailed token-level analysis of hallucination behaviors in Large Vision-Language Models (LVLMs). Unlike prior work that focuses on global or image-level statistics, the authors investigate hallucination phenomena at the patch- and token-level through cross-modal attention and feature alignment analysis. The study identifies two distinctive characteristics of hallucinated tokens: diffuse and spatially scattered attention distributions, and weak semantic alignment with any visual region. To quantify these findings, the authors introduce two novel measures—Patch-wise Attention Spatial Entropy and Patch-wise Cross-Modal Feature Similarity—and integrate them into a lightweight, interpretable hallucination detector. Experimental evaluation across multiple LVLM architectures demonstrates that the proposed approach achieves up to 90% accuracy in distinguishing hallucinated from grounded tokens.

**Strengths:**

This paper presents an interpretable analysis of hallucinations in multimodal models from the perspective of token-level representations. The authors’ introduction of spatial entropy and feature alignment as objective metrics is both intuitive and empirically validated, revealing consistent patterns between hallucinated and real tokens across architectures and layers. The proposed detector is computationally lightweight, explainable, and demonstrably effective, achieving state-of-the-art accuracy without relying on external supervision or large-scale retraining.

**Weaknesses:**

1. Does the training of a classifier heavily depend on the data distribution of the training set? If the training data differ significantly from the real-world data, will it cause the model to fail?

2. Although lightweight in computation compared to full model retraining, the approach still requires fine-grained token-level computation across layers and modes, which could be computationally intensive for large-scale deployment or streaming applications.

3. Lack of comparison with other state-of-the-art methods on public datasets, such as POPE[1] and MME[2].

4. Both this method and PATCH[3] adopt a visual patch. Please explain the main differences and the distinct starting points between the method proposed in this paper and that method.

[1] Evaluating Object Hallucination in Large Vision-Language Models

[2] MME: A Comprehensive Evaluation Benchmark for Multimodal Large Language Models

[3] From Pixels to Tokens: Revisiting Object Hallucinations in Large Vision-Language Models

**Questions:**

Please refer to Weaknesses

---

### Official Review · Reviewer_xSqC · 2025-10-30

**Soundness:** 2
**Presentation:** 3
**Contribution:** 3
**Rating:** 4
**Confidence:** 5

**Summary:**

The paper under review proposes two metrics for token wise hallucination detection for vision language models (VLM). First, the authors propose to exploit spatially distributed cross-attention of visual token patches and measure its entropy (after discarding some classes of lesser entropy) to obtain a heatmap. The authors observe that for non-hallucinated object tokens this entropy is focused in the visual region occupied by the object corresponding to the token, resulting in low entropy, while for hallucinate tokens the attention is diffuse and thus entropy is high. The second metric the authors design is a feature similarity of the language features with visual features over tokens. Here the authors extract k tokens with the highest (similarity?) score and average over these to obtain a second metric. Both metrics are tested for their capability to separate correct from hallucinated object tokens and achieve AUROC values around 80%. The tests are conducted on a self curated data set from 4000 COCO images  with ChatGPT 4.0 as arbitrary for correct vs hallucinated. Both metrics are thereafter combined into a single score using Random Forests, a FCNN or an XGBoost trained on sample data. The authors conduct comparisons over various VLM and compare themselves to recent hallucination detection baselines, finding considerable improvement over the competing models.

**Strengths:**

+ The paper deals with an important and timely topic
+ The proposed metrics seem to be new in the given context and show promising performance
+ Comparison with baselines shows considerable improvement over recent hallucination detection methods
+ The attention-entropy method results in images that can to some degree be interpreted by humans
+ The figures are instructive and the paper is well written and easy to follow

**Weaknesses:**

- The main weakness I see is the evaluation on a self-curated data set consisting out of 4000 images from COCO and ChatGPT as arbitrary. It has not become clear to me, why standard metrics like CHAIR are not evaluated. This  would help the comparison with existing methods a lot. Like this, it is even not described in detail how this subset of COCO is curated.
- In the same direction, only a single benchmark is evaluated. Others, like POPE, are just left our even though this in the meantime has become a standard benchmark to evaluate.
- The spatial attention entropy metric is not completely new, the authors might like to cite [1] for usage in a different, but related context.
- Details on the training of the XGBoost, RF and FCNN are missing - e.g. what was the data split? Was it separate from the test data?
- Sometimes, the paper has a tendency toward 'colorful language' where the precise scientific content is difficult to grasp, starting with "The devil wears tokens", "interactions between patch tokens", "fine grained statistics on the patch level", "structural dynamics within the model" etc.
- Ablation studies, except for the different classifiers, are largely missing, e.g. with respect to the cut off parameter in the attention masks or the patch features.
- A study on the obtainable effects from hallucination detection, e.g. via nucleus sampling or contrastive decoding is missing.

My main concern is the insufficient evaluation. If this can be improved and the findings are confirmed throughout, I would find this an interesting paper.

[1] Krzysztof Lis, Matthias Rottmann, Annika Mütze, Sina Honari, Pascal Fua, Mathieu Salzmann, AttEntropy: [....], BMVC 2024

**Questions:**

* I expect more diffuse attention entropy maps for larger objects - has a study of the efficiency of the attention entropy metric as depending on the object size, e.g. available via the COCO segmentation masks, been conducted?
* Is your data curation over COCO randomized or have other criteria played a role. Does your subset e.g. lean towards smaller objects?
* Which are the data sets and the parameter settings for the classifier models?

---

### Official Review · Reviewer_T71p · 2025-10-31

**Soundness:** 3
**Presentation:** 3
**Contribution:** 2
**Rating:** 4
**Confidence:** 4

**Summary:**

This paper focuses on addressing the object hallucination issue in Large Vision-Language Models (LVLMs).

**Strengths:**

Innovative Fine-Grained Analysis Perspective: The paper breaks away from the limitations of traditional global-level hallucination analysis and pioneers a patch-token level investigation.

**Weaknesses:**

1. Limited Generalization to Non-Object Hallucinations: The study exclusively focuses on object hallucination in LVLMs, ignoring other types of hallucinations such as attribute hallucinations or relational hallucinations. This limits the general applicability of the proposed method to broader hallucination scenarios in LVLMs.

2. Potential Sensitivity to Hyperparameters: The detector relies on several hyperparameters, such as the top-x% of patch responses (x=10) for attention entropy calculation and the number of top-k layers (k=10) for feature aggregation. The paper only mentions that these hyperparameters are empirically determined but lacks a systematic analysis of their impact on detection performance. This makes it unclear how the detector would perform when hyperparameters are adjusted or applied to different LVLMs with varying architectures.

3. Lack of Exploration into Hallucination Mitigation: While the paper effectively detects hallucinations, it does not extend its insights to hallucination mitigation.

4. Dependence on GPT-4o for Dataset Labeling: The classification of true and hallucinated object tokens in the dataset relies on GPT-4o, which may introduce annotation biases. The paper does not validate the accuracy of GPT-4o's labels (e.g., through human annotation verification) or discuss potential errors in label assignment, which could affect the reliability of the experimental results and the training of the hallucination detector.

5. Lack of some Related Works: How does this paper differ from DHCP, and how do their performances compare? They all utilize the attention of LVLMs for hallucination detection.

DHCP: Detecting Hallucinations by Cross-modal Attention Pattern in Large Vision-Language Models, uploaded to Arxiv in 2024.

**Questions:**

See weaknesses.

---

### Official Review · Reviewer_nPEp · 2025-11-01

**Soundness:** 2
**Presentation:** 3
**Contribution:** 2
**Rating:** 4
**Confidence:** 4

**Summary:**

The paper investigates object hallucination in large vision–language models at the token level and argues that hallucinated object words exhibit two consistent structural patterns: their visual attention is spatially diffuse with higher entropy, and their cross-modal similarity to any image patch is weak. Building on these observations, the authors construct a lightweight detector that aggregates layerwise attention-entropy and patch-similarity features and show that it outperforms prior token-level hallucination detectors (e.g., MetaToken, HalLoc, SVAR) across several LVLM architectures, suggesting that these two signals capture model-agnostic properties of ungrounded generations.

**Strengths:**

1. The paper makes the empirical observation that real-object tokens localize attention in a few contiguous patch components, whereas hallucinated tokens spread attention and often hit attention sinks.
2. Results on LLaVA, InternVL, MiniGPT-4, and GPT-4V suggest the patterns are not tied to a single architecture

**Weaknesses:**

1. The paper proposes a hallucination detector, yet all quantitative comparisons are conducted on the authors’ own COCO–GPT-4o curated token-level dataset. This makes it difficult to assess how much of the reported gain over other methods would persist on established hallucination benchmarks (e.g. CHAIR, POPE, ROPE) or on publicly available multi-image/object settings.
2. The analysis and experiments are conducted around an image-description style interaction, and the paper does not show whether the proposed token-level signals remain equally discriminative in more diverse settings.
3. Although the paper describes the detector as simple and lightweight, the actual feature pipeline requires extracting attention maps and visual patch embeddings from all transformer layers, running per-layer component filtering and attention-sink suppression, and computing patch-level similarities. This is considerably more expensive than baselines such as MetaToken or SVAR, and may limit applicability in online or long-form generation settings. A complexity / runtime report would help quantify this overhead.

**Questions:**

1. Can you report results when the detector is plugged into existing hallucination benchmark pipelines so we can see whether the relative gains over MetaToken / HalLoc / SVAR still hold in a standard setting?
2. How sensitive is the detector to decoding changes? Since these will change token trajectories, it would be useful to know whether your features remain stable.
3. Is there a way to utilize your analysis into mitigation methods that could reduce hallucination on standard benchmarks?

---

### Note · Authors · 2025-11-13

**Comment:**

We appreciate the reviewers' and ACs' efforts in reviewing our work. We will incorporate the feedback and revise the manuscript to clarify and better highlight our technical novelty.

Best regards,
On behalf of the authors

**Withdrawal Confirmation:**

I have read and agree with the venue's withdrawal policy on behalf of myself and my co-authors.